# SYN-View: A Phylogeny-Based Synteny Exploration Tool for the Identification of Gene Clusters Linked to Antibiotic Resistance

**DOI:** 10.3390/molecules26010144

**Published:** 2020-12-31

**Authors:** Jason Stahlecker, Erik Mingyar, Nadine Ziemert, Mehmet Direnç Mungan

**Affiliations:** 1Interfaculty Institute of Microbiology and Infection Medicine, University of Tübingen, Auf der Morgenstelle 28, 72076 Tübingen, Germany; wilhelm-jason.stahlecker@student.uni-tuebingen.de (J.S.); erik.mingyar@uni-tuebingen.de (E.M.); nadine.ziemert@uni-tuebingen.de (N.Z.); 2German Centre for Infection Research (DZIF), Partner Site Tübingen, 38124 Tübingen, Germany

**Keywords:** biosynthetic gene clusters, natural products, genome mining, antibiotic resistance

## Abstract

The development of new antibacterial drugs has become one of the most important tasks of the century in order to overcome the posing threat of drug resistance in pathogenic bacteria. Many antibiotics originate from natural products produced by various microorganisms. Over the last decades, bioinformatical approaches have facilitated the discovery and characterization of these small compounds using genome mining methodologies. A key part of this process is the identification of the most promising biosynthetic gene clusters (BGCs), which encode novel natural products. In 2017, the Antibiotic Resistant Target Seeker (ARTS) was developed in order to enable an automated target-directed genome mining approach. ARTS identifies possible resistant target genes within antibiotic gene clusters, in order to detect promising BGCs encoding antibiotics with novel modes of action. Although ARTS can predict promising targets based on multiple criteria, it provides little information about the cluster structures of possible resistant genes. Here, we present SYN-view. Based on a phylogenetic approach, SYN-view allows for easy comparison of gene clusters of interest and distinguishing genes with regular housekeeping functions from genes functioning as antibiotic resistant targets. Our aim is to implement our proposed method into the ARTS web-server, further improving the target-directed genome mining strategy of the ARTS pipeline.

## 1. Introduction

With the increasing number of drug-resistant bacteria, antimicrobial resistance has become a global health threat [1]. As the number of approved drugs have been decreasing over the past few decades, finding new compounds to feed the antibiotic discovery pipeline has become a crucial task [2]. Most of the antibiotics are derived from secondary metabolites (SMs) produced by fungal and bacterial organisms [3]. Many of these so-called natural products were found by labor-intensive methods such as screening biological samples for desired bioactivities. However, these traditional methods have been losing their efficiency, due to their high rediscovery rates [4]. Ever since the cost of DNA sequencing technologies has decreased substantially, in silico methods such as genome mining have gained an increased amount of popularity among researchers [5,6]. As a result, a number of computational tools such as antiSMASH [7] and PRISM [8] have been developed, in order to detect gene clusters encoding for natural products. The main approach of these tools is the identification of locally clustered groups of genes called biosynthetic gene clusters (BGCs), which are in conjunction responsible for the synthesis of secondary metabolites [9]. Using those BGC prediction tools, a large number of BGCs have been deposited in public databases. The newest version of Atlas of Biosynthetic Gene Clusters (IMG-ABC) [10], the largest database containing predicted BGCs, contains roughly 400,000 clusters, from which less than 1% have been experimentally verified. This large discrepancy emphasizes the need for new and updated tools as well as the importance of prioritization of predicted BGCs for downstream processes. In order to address this issue, in 2017, Alanjary et al. developed the Antibiotic Resistant Target Seeker (ARTS) [11] to detect most promising BGCs with potential new modes of action by automating the resistance based genome mining technique also called target directed genome mining. This approach is based on the notion that the antibiotic-producing bacteria have to be resistant to their own products [12]. Resistance genes can be encoded within the BGC of the respective compound. Additionally, in case of a resistance mechanism that is provided by a resistant target, this kind of genome mining method not only provides insights into the mode of action of the encoded antibiotics, but in turn also allows screening BGCs for natural products with promising and putatively novel targets [13]. ARTS links essential housekeeping genes to evolution driven events such as duplication, horizontal gene transfer (HGT), or co-localization within the BGC, which have been extensively shown to be the key processes in target-based strategies [14,15,16]. Although ARTS rapidly screens essential genes of an entire genome, the number of potential resistant targets can become quite large, especially when the BGC boundaries are set too far. In such cases, the distinction of a resistance gene and a regular housekeeping gene is hard to make. As stated by O’Neill and his colleagues, inferring such distinctions may be possible, by comparing gene ortholog neighbors of the putative resistance genes and the context of the clusters they lie within in related organisms. Regular housekeeping genes often show synteny in their cluster structure, whereas the resistant target genes within antibiotic gene clusters are often only randomly present in closely related taxa [17]. Following up on this hypothesis, we analyzed the novobiocin producer *Streptomyces niveus* NCIMB 11891, with duplicated *gyrB* gene as known self-resistance mechanism, yielding a large number of false positives by an initial ARTS search shown in the first ARTS paper [11]. Visualized in Figure 1, the comparison of the neighborhoods of gene of interest (NGIs) to the NGIs from closely related organisms, clearly shows that the neighborhood of the housekeeping gene is almost identical, whereas the resistant target gene shows no orthologous genes in the neighborhood.

While the housekeeping genes play an important role in target-directed genome mining approaches and BGC prioritization, the context of the gene neighborhood has not yet been focused on. In order to address this issue, here we introduce SYN-view, for further improvement of prioritization of the BGCs, based on a self-resistance approach. With the aid of phylogenetic methods such as autoMLST [18], which provides a high-resolution species tree of a strain of interest, SYN-view compares NGIs, based on user-provided target genes to homologous NGIs from closest relatives. Unlike other tools such as MultiGeneBlast [19], which blasts a complete cluster to a specific database to find similar clusters, our pipeline aims to distinguish a potential target resistance gene from regular housekeeping genes, by rapidly comparing NGIs from closely related taxa.

## 2. Results and Discussion

Here, we present SYN-view, an easy-to-use pipeline in order to make rapid comparison of NGIs and provide an additional way to detect putative novel antibiotic resistant targets. SYN-view allows for easy to interpret visualizations of NGIs in order to distinguish genes of interest with different functions. Using an external tool such as autoMSLT [18], SYN-view uses homology search tools to find the input protein and its surrounding genes from closest taxa, in order to perform a synteny search for easy detection of unique gene cluster structures. SYN-view can be easily installed using conda packages [20] and is publicly available at https://bitbucket.org/jstahlecker/syn-view/. An overview of the workflow is illustrated in Figure 2.

### 2.1. Positive Controls

For the proof of concept of our proposed method, first we examined bacterial strains reported for antibiotic production with known resistance mechanisms shown in Table 1, to test if there is a significant difference between NGI structures of regular housekeeping genes and genes responsible for self-resistance. Results suggested that when the resistance mechanism includes a duplication event, difference in respective NGIs can be easily recognized. In certain cases where resistance genes have been mutated instead of duplicated (Table 1, *A. mediteranei* S699, *rpoB*), differences in NGIs could not be observed. Nevertheless, it would be possible to detect a difference in NGIs even if there is no duplication of self resistance genes but if they are unique to a certain bacterial genome. All of the corresponding results are visualized in detail in the Supplementary Results.

### 2.2. SYN-View as a Complementary Method

In order to prove that SYN-view can further improve the current ARTS pipeline as a complementary method, we employed a final test case where ARTS could not find hits for a known self resistance mechanism. As stated in the first ARTS paper, 23S rRNA methyltransferase, which confers resistance for Avilamycin, was undetected by hmmsearch due to its short sequence length and low homology score. As HMMs are dependent on profiles built from multiple sequence alignment [22], it may fail to represent sequences which are not fully reflecting specific domains characterized from respective proteins. For such cases, SYN-view supports homology search using blastp algorithm, which makes it possible for users to analyze shorter sequences or proteins without an accurate HMM model. As shown in Appendix A, the synteny among closest relatives of the NGI of 23S rRNA methyltransferase, conferring self-resistance, is significantly different than the NGI with regular housekeeping function.

## 3. Materials and Methods

### Input Options and Workflow

An overview of the workflow is illustrated in Figure 2. First, SYN-view needs an annotated genome file in GenBank format (gbff, gbk). Additionally, an HMM or protein fasta file for a gene/protein is required, which is used to either run hmmsearch [23] or blastp [23], against the input genome to find similar proteins. SYN-view uses default cut-off values for hmmsearch and blastp algorithms, which can be redefined by the user. Using Biopython [24], the input genome is parsed and per hit, a query NGI is created based on the proximity of the respective hit. By default, this proximity setting is three surrounding genes in both sides of the gene of interest; however, it can be changed to decrease/increase the size of the NGI. Finally, close relatives of the input genome must be set, for the synteny search. For this purpose, the user can either provide the result file of an autoMLST job (mash_distances.txt, recommended) or provide a custom folder with specified genomes in GenBank format. If an autoMLST result is provided, the 10 closest organisms are, by default, downloaded from NCBIs RefSeq database [25]. As stated earlier, increasing the number of closest organisms would also increase the quality of the result. For the purposes of speed, it was set as 10 for default but can be changed via command line arguments. After downloading genomes, the next part of the SYN-view pipeline is detecting the input protein sequences from given genome, creating query NGIs. Afterwards, a database is created, containing only the NGIs from the closest relatives based on the input protein. The NGIs of the input are then blasted against the database and the NGI hits are scored by cumulative blast bit score. Higher bit scores suggest higher sequence similarity, while being independent of the database size. Therefore, summing over all individual bit scores of a NGI gives an indication of the sequence similarity of the whole NGI with respect to the query. In the results folder, all NGI hits per query can be analyzed using the corresponding visualization as an svg file as explained in Supplementary Results and can be compared to other hits using a standard web browser (Figure 1). The color coding makes it easier to identify similar hits to unique gene cluster structures. Same color indicates similarity to the query protein, while white suggests no hits, with the exception of being white colored in the query NGI, which indicates that the protein does not have a defined translated sequence.

## 4. Conclusions

With SYN-view, we developed a program that allows a rapid and easy to interpret overview about the gene neighborhoods of genes of interest. This can be used as an additional criterium to detect putative antibiotic resistant targets. However, SYN-view can also be used for the exploration of cluster formations of specific genes in phylogenetically similar bacteria. A preceding prioritization of genes of interest such as an ARTS run is recommended, as both tools utilize self-resistance. As it is impossible to identify resistance genes based on a single criterion, SYN-view is meant to be used as a complementary tool to help researchers in their efforts for the prioritization of their targets. As the genomic content of a NGI is specific for different cases, it is incumbent on the user to further analyze results. In order to further automate this workflow and increase efficiency our aim is to implement this functionality in ARTS web-server.

## Figures and Tables

**Figure 1 molecules-26-00144-f001:**
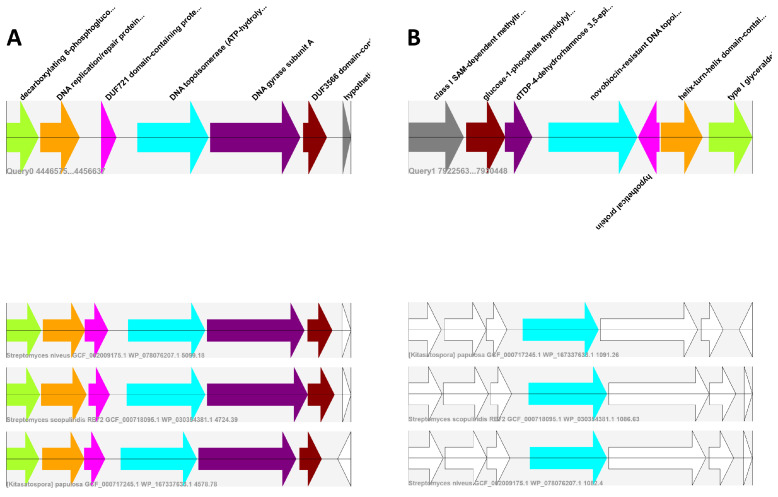
Exemplary result of SYN-view. The figure shows two alignments of NGIs throughout the closest relatives of *Streptomyces nivues* NCIMB 11891 (Table 1). Note that for a clear comparison, only two NGI alignments are shown, while three were found (Supplementary Data). (**A**) NGI of DNA topoisomerase which is regularly observed in close relatives with the structure of the NGI is well conserved. (**B**) The NGI of the duplicated, resistant *gyrB* is unique to the antibiotic producing strain and can easily be distinguished.

**Figure 2 molecules-26-00144-f002:**
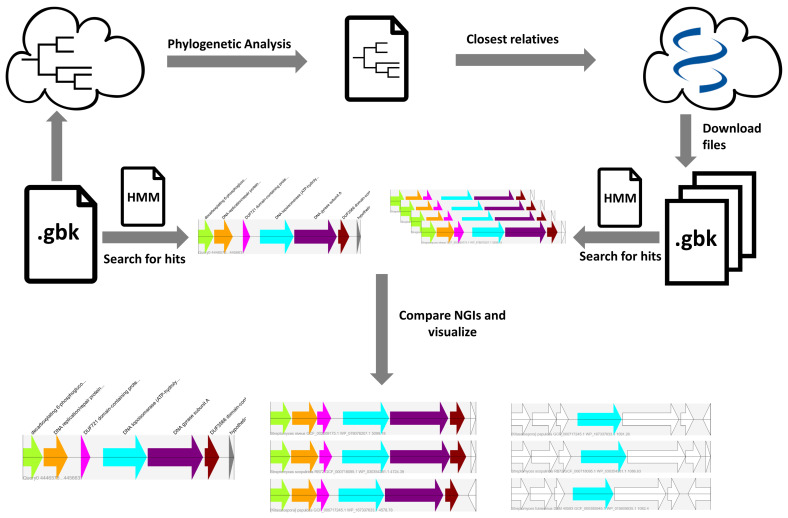
Schematic workflow. A phylogeny file needs to be created using autoMLST or an appropriate folder must be specified. Based on that, the 10 closest relatives are downloaded from the NCBI refseq database. Using an appropirate hmm or protein fasta file, NGIs are created, scored, and sorted. Finally, the results are saved as an svg file.

**Table 1 molecules-26-00144-t001:** SYN-view analysis of example antibiotic producing strains with identified self-resistance genes. For comparison, respective ARTS hits are also provided from previous papers [11,21] (D: Duplication, B: BGC proximity, R: Resistance, P: Phylogeny). “Search Type” column indicates how the search was performed: H stands for HMM mode while B stands for blastp and the following indicates the corresponding TIGRFAM model and gene accession number, respectively. Easily identifiable differences are denoted as “Yes”, if no difference is visible marked as “No”.

Organism	Resistance Gene	Search Type	ARTS Hits	Identifiable
*Streptomyces niveus* NCIMB 11891	*gyrB*	H: TIGR01059	D,B,R,P	Yes
*Streptomyces roseochromogenes* DS 12.976	*gyrB*	H: TIGR01059	D,B,R,P	Yes
*Burkholderia thailandensis* E264	*accA*	H: TIGR00513	D,B,R,P	Yes ^a^
*Salinospora tropica* CNB-440	beta-proteasome subunit	H: TIGR03690	D,B,R,P	Yes ^a^
*Myxococcus xanthus* DK 1622	*lspa*: signal peptidase II	H: TIGR00077	D,B,P	Yes
*Bacillus cereus* ATCC 14579	duplicated RL11	H: TIGR01632	D,P	Yes
*Nordica farnica* IFM 10152	*rpoB*	H: TIGR02013	D	Yes
*Agrobacterium radiobacter* K84	Leu-tRNA synthase	H: TIGR00396	D,P	Yes
*Streptomyces viridochromogenes* Tue57	23S rRNA methyltransferase	B: AAG32066.1	No Hits	Yes
*Amycolatopsis mediterranei* S699	*rpoB*	H: TIGR02013	R	No

^a^ A difference was better observed after using 50 rather than the default 10 closest genomes.

## Data Availability

Data is contained within the article or supplementary material.

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
