# Peer review of "SYN-View: A Phylogeny-Based Synteny Exploration Tool for the Identification of Gene Clusters Linked to Antibiotic Resistance"

_molecules, 2020, doi:10.3390/molecules26010144_

Round 1

Reviewer 1 Report

The manuscript presents a phylogeny-based bioinformatic tool, namely SYN-view, which is used to make rapid comparison of the neighborhoods of gene of interest (NGIs) and distinguish resistance genes from regular housekeeping genes in the target directed genome mining strategy (TDGM) of the ARTS pipeline. The SYN-view program is very useful in genome mining as the introduction of SYN-view into TDGM improves the accuracy of the resistance gene prediction and is complementary to ARTS for the prioritization of target biosynthetic gene clusters. I think this manuscript is worthy to publish in Molecules and the tool will attract more attention from researchers in the fields of natural product genome mining and genomic analysis. My only concern is Figure 1, the annotation of which looks too faint and hard to read. The authors should increase the size of the fonts for the annotation above each gene and the title below each gene cluster.

Author Response

We thank the reviewer for the constructive feedback and to address the raised concern, we increased the font sizes and changed it to bold style for easier viewing both in the papers figures and the output figures from SYN-view.

Reviewer 2 Report

Stahlecker and colleagues have developed a new tool for analysing the clusters of genes adjacent to potential antibiotic resistance genes within biosynthetic gene clusters by looking at the same gene and its adjacent genes in phylogenetically related strains. This tool, SYN-view is meant to act as a module within an existing web server for the Antibiotic Resistant Target Seeker (ARTS).

The SYN-view Tool, I believe is a very useful addition to a useful web based pipeline, that can also be downloaded and used locally. The manuscript is well written and I fully support its publication. I would like to congratulate the authors for producing this work and their team’s ongoing contribution to the field.

My recommendations for improvement of the manuscript include;

-Line 31-32: Change sentence to “Using those BGC prediction tools, a large amount of BGCs have been deposited in public databases.”

-Line 41: Change to “resistance mechanism”.

-Line 85: Change “tpoB” to “rpoB”.

-Table 1 contains the results of an example run of SYN-view. The data for this is included in the supplementary data. However similar to Figure 1 and Figure S1, I think it would be useful for readers if the authors also displayed the crucial results in easy to view figures of the top alignments that lead the authors to conclude from the output if there are easily identifiable differences.

Author Response

We thank the reviewer for the thorough comments and analysis of the paper.
The minor changes regarding the writing have been made and the whole sentence change is highlighted in red for line 31-32.
For the mentioned results in table 1, we added all the corresponding comparative figures to supplementary pdf file as well.